# Acoustic Atomization-Induced Pumping Based on a Vibrating Sharp-Tip Capillary

**DOI:** 10.3390/mi14061212

**Published:** 2023-06-08

**Authors:** Balapuwaduge Lihini Mendis, Ziyi He, Xiaojun Li, Jing Wang, Chong Li, Peng Li

**Affiliations:** 1C. Eugene Bennett Department of Chemistry, West Virginia University, Morgantown, WV 26506, USA; bcm0021@mix.wvu.edu (B.L.M.); wendyxjl@hotmail.com (X.L.); jw0156@mix.wvu.edu (J.W.); chongli52900@gmail.com (C.L.); 2College of Veterinary Medicine, Huazhong Agricultural University, Wuhan 430070, China; zyhe@mail.hzau.edu.cn

**Keywords:** acoustofluidic pump, atomization-based pumping, vibrating sharp-tip

## Abstract

Pumping is an essential component in many microfluidic applications. Developing simple, small-footprint, and flexible pumping methods is of great importance to achieve truly lab-on-a-chip systems. Here, we report a novel acoustic pump based on the atomization effect induced by a vibrating sharp-tip capillary. As the liquid is atomized by the vibrating capillary, negative pressure is generated to drive the movement of fluid without the need to fabricate special microstructures or use special channel materials. We studied the influence of the frequency, input power, internal diameter (ID) of the capillary tip, and liquid viscosity on the pumping flow rate. By adjusting the ID of the capillary from 30 µm to 80 µm and the power input from 1 V_pp_ to 5 V_pp_, a flow rate range of 3 to 520 µL/min can be achieved. We also demonstrated the simultaneous operation of two pumps to generate parallel flow with a tunable flow rate ratio. Finally, the capability of performing complex pumping sequences was demonstrated by performing a bead-based ELISA in a 3D-printed microdevice.

## 1. Introduction

For most microfluidic applications, fluid pumps are essential to achieve various fluid manipulations. Syringe pumps [1,2,3] or pneumatic pumps [4,5] are the most used pumps due to their wide availability and accurate flow rate control. However, the drawback of these pumps is their bulkiness, which does not match the scale of most microfluidic devices and hinders the development of lab-on-a-chip systems with complex functionality. Therefore, developing miniaturized pumping methods is of great importance to further expand the application scope of microfluidic devices.

To date, a lot of efforts have been devoted to developing micropumps that are compatible with microfluidic devices. Based on the driving mechanisms, there are two types of micropumps: passive pumps [6,7] and active pumps [8,9,10]. Passive pumps are based on surface tension or gravity to move fluids in a microchannel, which do not need peripheral equipment for the operations [7]. These pumping methods are suitable for simple applications that do not require complex fluid manipulation. For applications that need precise and on-demand fluid control, active pumps are preferred. Active pumps employ external actuation mechanisms to drive the fluid. Unlike passive pumps, active pumps offer better flexibility in fluid manipulation. Typical active pumps include optically driven pumps [11,12], electric pumps [13,14], magnetic pumps [9,10], pneumatic membrane pumps, and acoustic pumps [15,16,17,18,19,20,21,22,23,24,25,26,27,28,29,30,31,32,33,34].

Among these pumping mechanisms, acoustic pumps have become increasingly important in recent years due to their small footprint, simple setup, and flexible operation. Early acoustic pumps induce deformations on channel walls, thereby driving the fluid movement inside a microchannel [31]. Although this strategy is effective, its energy efficiency and the requirement of flexible channel walls limit its broad application. Travelling surface acoustic wave (TSAW) induced body force and expansion force on liquid have also been used for fluid pumping. Tao et al. [15] reported TSAW-based pumping by applying RF signals to microfabricated interdigital transducers (IDTs). The device was modified with a hydrophobic coating to improve the pumping performance. The authors were able to achieve a pumping flow rate in the range of 0.1 to 0.2 μL/min with a power consumption of 2–7 W. More recently, acoustic pumps based on acoustic streaming have also been reported [16,33,35]. Acoustic streaming is the stable fluid flow induced by the dissipation of acoustic energy to bulk fluid [34,36,37]. By controlling the direction and pattern of acoustic streaming, the unidirectional movement of a liquid can be achieved. It has been reported that sharp-edge structures can induce a strong acoustic streaming vortex for fluid pumping. Huang et al. [16] reported the first sharp-edge-based pumping device by fabricating PDMS sharp-edge structures with a tilt angle to the channel wall, which breaks the symmetry of the sharp-edge induced acoustic streaming, leading unidirectional fluid low. This device achieved flow rates up to 8 μL/min at an input voltage of 50 V. Pavlic et al. [38] reported a microfluidic chip that has acoustically actuated sharp edges at the tip of the channel opening that induces a flow in the microchannel made of silicon and glass. The oscillations of the sharp edges were able to induce a net fluid flow that reached up to 4.1 μL/min. In addition, SAW-induced acoustic streaming can also be utilized for fluid pumping. Wu et al. [39] reported C-shaped IDTs to generate localized acoustic streaming for fluid pumping. This method achieved flow rates in the range of 18.5 nL/min to 41.5 nL/min with a power consumption of 2–6 W. Although acoustic streaming-based pumps are simple and have a small footprint, these pumps make it difficult to achieve complex fluid manipulations with multiple reagents due to the non-selective excitation of bulk acoustic vibration.

In this work, we present a novel acoustic pump that is based on the acoustic atomization effect. This method utilizes the fluid atomization effect generated by a vibrating sharp-tip capillary, which has been used as an ionization source for mass spectrometry analysis. In this work, we demonstrated that this phenomenon can be used as an efficient pumping method. As the fluid is atomized at the capillary tip, a negative pressure is generated to the bulk fluid, thereby creating unidirectional fluid flow. It has been reported that SAW-induced atomization can induce continuous fluid flow [40,41,42]. However, the requirement of a SAW substrate and microfabricated electrodes limited their utility as a generic microfluidic pump. The present method allows the independent control of each pumping unit, and the pumping performance is independent of the channel material and microstructures, which can be adopted by any microfluidic system. We studied the relationship between the flow rates and various design and operational parameters, including the capillary geometry, input frequency, and amplitude. This pump achieved a flow rate range of 3 to 520 μL/min. We demonstrated parallel flow streams in a microchannel by operating two pumps simultaneously. Finally, complex fluid operations were demonstrated by performing a complete bead-based ELISA protocol in a 3D-printed microdevice.

## 2. Materials and Methods

### 2.1. Reagents and Materials

Amine-modified polystyrene fluorescent orange latex beads (2.0 μm) were bought from Sigma Aldrich (St. Louis, MI, USA). The microparticle solution was prepared by mixing 5 μL of the microsphere stock solution into 495 μL of DI water. Rhodamine B powder was purchased from Sigma Aldrich. Rhodamine B fluorescent dye solution was prepared by mixing 1 mg of Rhodamine B powder with 10 mL of DI water, which was vortexed for complete dissolution. The water was purified using a Millipore purification system (Bedford, MA, USA). Magnetic microparticles (5 μm) were bought from Sigma Aldrich. Streptavidin solution, QuantaRed-enhanced chemifluorescent HRP substrate, and phosphate-buffered saline (PBS) were bought from Thermo Fisher Scientific (Waltham, MA, USA). A Quantikine ELISA commercial Activin A Immunoassay Kit was purchased from R&D Systems, Inc., Minneapolis, MN, USA. Glycerol was purchased from Sigma Aldrich. Solutions of different viscosities were prepared by diluting 50.00, 100.00, 150.00, and 200.00 μL of glycerol in DI water into a final volume of 600 μL. Solutions of 4 different viscosities of 0.00115, 0.00153, 0.00209, and 0.00297 Ns/m^2^ (1.2 cP, 1.5 cP, 2.1 cP, and 3.0 cP, respectively) were made for the viscosity experiment.

### 2.2. Device Design and Fabrication

The acoustically activated pulled-tip glass capillary pump was fabricated based on our previous reports [43,44,45,46,47]. Briefly, a piezoelectric transducer (7BB-27-4L0, Murata, Kyoto, Japan) and a pulled-tip glass capillary were glued to the opposite ends of a microscope cover glass slide (24 × 60 mm, purchased from VWR). The piezoelectric transducer was fixed to the glass slide using epoxy glue (5 min epoxy, Devcon). The pulled-tip glass capillary was made by pulling 0.4 mm ID capillary tubes purchased from Drummond Scientific (Drummond Scientific, Broomall, PA, USA) using a laser-based micropipette puller (P-2000, Sutter Instrument, Novato, CA, USA). The pulled-tip glass capillaries with their IDs in the range of 20 to 80 μm were fixed onto the opposite edge of the glass slide from the piezoelectric transducer using glass glue (Loctite Rocky Hill, CT, USA) with a 30-degree angle between the capillary and the shorter side of the glass slide and with a distance to the corner of ~5 mm, as noted in Figure 1a. The angles and position of the capillary were optimized based on our previous work [43].

All the 3D-printed microfluidic devices were first designed using SolidWorks. Then, the devices were printed using a Phrozen Sonic 4K resin 3D printer with an X-Y resolution of 35 μm. The UV light source wavelength of the printer is 405 nm. The Nanoclear resin was purchased from Funtodo, and the SC-801 resin was purchased from Phrozen. After the printing was complete, the printed device was removed from the build platform and washed with Isopropanol to dissolve the excess resin inside the channels. After a post-exposure process under UV (365 nm) for 5 min, the device was glued to a glass slide (VWR, Radnor, PA, USA) using epoxy glue (5-min epoxy, Devcon) to enhance the transparency of the device. The inlets and outlets of the device were connected to 0.034″ I.D. × 0.060″ O.D tubing purchased from Scientific Commodities. The device used for the on-chip bead-based ELISA comprises a well in the middle of the device to house the magnetic beads with a removable cover fixed to it. A magnet was used to immobilize the magnetic microparticles during the washing steps of the ELISA. The PDMS device was fabricated following standard soft lithography procedures, as reported in our previous work [48].

### 2.3. Pump Operation

The vibrating sharp-tip capillary pump was connected to the 3D-printed microdevice via tubing. The whole system was then filled with water or other solutions manually. Then, the outlet tubing was dipped in the reservoir containing the water or other desired solutions. The pumping was then activated and controlled using a Tektronix function generator (AFG1062) connected to an amplifier (LZY-22+, Mini-Circuits, Brooklyn, NY, USA). The typical power input range is 1–5 V_pp_, and the working frequency range is 90–100 kHz. During the pump operation, the flow rate was adjusted by changing the input power. To study the pumping flow rate, fluorescent particles were introduced into the microchannel. The movement of the particles was recorded using an epifluorescence microscope (Olympus IX 73) and an sCMOS camera (Hamamatsu Orca-Flash 4.0 LT+).

### 2.4. On-Chip Magnetic Bead-Based ELISA

A total of 2 μL of the 1 mg/mL streptavidin solution was mixed with 2.5 μL of 5 μm particle-sized magnetic beads and 998 μL of PBS. Then, the solution was incubated for 3 h in the rotating stirring machine. Then, the microparticles were washed 4 times with PBS (200 μL at each step). The streptavidin-coated beads were then resuspended in 5 μL of PBS. A total of 200 μL of the biotinylated capture antibody solution was added to the bead solution and incubated at room temperature for 15 min, followed by 3 washing steps with 200 μL of washing buffer at each step. Then, the bead solution was mixed with 500 μL of 3% BSA and incubated for 45 min under room temperature, followed by 3 washing steps with 200 μL of washing buffer at each step. Then, the beads were resuspended in 5 μL of PBS and introduced into the 3D-printed microdevice.

Inside the well of the sample device, 200 μL of the sample mix (a mixture of 100 μL of assay diluent and 100 μL of 1000 pg/mL sample solution) was gradually pumped through the well using the stop-flow method throughout the incubation time of 3 h. For the control, 200 μL of only the assay diluent was pumped. Then, the wash buffer was pumped 3 times across the well using 200 μL at each step. During the reagent introduction steps, the beads were held in place using a magnet that was placed under the device across from the well. In addition, during the incubation times, the magnet was removed. Then, 200 μL of the detection antibody (Activin A conjugate) solution was pumped and incubated for 1 h, followed by 4 washing steps using 200 μL of wash buffer at each step. Then, the substrate mix was prepared by mixing 200 μL each from the QuantaRed stable peroxide solution and enhancer solution with 4 μL of QuantaRed ADHP concentrate. The substrate mixture was pumped through the sample and control devices (200 μL each) and incubated for 15 min before measuring the fluorescence intensity.

## 3. Results and Discussion

### 3.1. Operation Principle

In our previous works, we reported that a liquid-filled sharp-tip capillary can generate small liquid droplets, which can serve as an ionization source mass spectrometry analysis, termed vibrating sharp-edge spray ionization (VSSI) [43,44,45,46,47]. Typically, in VSSI mass spectrometry experiments, a syringe pump is used to ensure continuous sample loading. We also observed that as droplets are generated from the vibrating capillary, it can also induce fluid flow to sustain the ionization process. Here, we investigated whether this phenomenon could generate sufficient pressure to serve as an efficient pump for microfluidic devices. The pumping unit consists of a piezoelectric transducer, a glass slide, and a pulled-tip capillary (Figure 1a). The piezoelectric transducer has a diameter of 27 mm, whereas the glass slide is 25 × 60 mm. The capillary tip was pulled to an ID of 30 μm using a laser puller. After assembling the device, a strong plume was observed when an RF signal was applied to the piezoelectric transducer (Figure 1b and Appendix A).

The microdevice tested in this study was fabricated using 3D printing. Based on the pumping mechanism of the present method, it is compatible with any microdevices regardless of its microfabrication methods or channel material. Here, we chose a 3D printed device for testing as other acoustic pumps have not been applied to 3D printed devices due to the limitation of the printing resolution and channel material. The testing device has a main fluidic channel of 900 μm × 950 μm × 1.5 cm (width × height × length), one outlet is connected to the pumping unit, and one inlet is connected to a reagent reservoir (Figure 1c). It should be noted that the pulled-tip capillary was covered with a small tube for collecting aerosols. The whole fluidic channel and the pumping unit were first filled with water, and the outlet was connected to the reagent reservoir with Rhodamine B. When applying an RF signal of 94 kHz and 8 V_pp_ to the piezoelectric transducer, a strong plume was observed at the tip of the capillary along with the movement of the Rhodamine B solution through the channel (Figure 2, Appendix A). The flow rate was estimated to be ~200 μL/min. This result indicated the feasibility of using the vibrating sharp tip as a pumping unit for microfluidic devices.

### 3.2. Characterization of Important Operational and Design Parameters on Pumping Performance

Next, we examined the factors that may affect the pumping flow rate. In this study, the flow rate was calculated by measuring the flow velocity of 2 μm fluorescent orange beads in the fluidic channel using a cross-section of 900 μm × 950 μm. We first studied the impact of the frequency on the pumping performance. The working frequency of the present system is dependent on the resonance frequency of the transducer, the device setup and its resonance frequency, and the damping in the system. We scanned the frequency from 93 kHz to 99 kHz. As shown in Figure 3a, the pump can work under a broad range of frequencies. In this work, we chose the frequency that generated the highest flow rate under the same power input as the optimal working frequency for the sharp-tip pump.

We also examined the relationship between the power input and the pumping flow rate. We tested a capillary with a tip ID of ~30 μm at a frequency of 95 kHz. As shown in Figure 3b, the flow rates increase with an increase in the applied power. The relationship between the flow rate and the input voltage is logarithmic instead of linear. We observed that the flow rate increased from 5 μL/min to 60 μL/min when the input voltage increased from 1 V_pp_ to 2 V_pp_. Further increasing the voltage from 2 V_pp_ to 5 V_pp_ only increased the flow rate from 60 μL/min to 85 μL/min. To control the flow rate of the sharp-tip pump, a calibration curve needs to be established. It should also be noted that the sharp-tip pump has an onset voltage for the pumping phenomenon. For the capillary tested here, the onset voltage was ~0.9 V_pp_. For voltages below this value, no atomization effect was observed, leading to no pumping for the microfluidic device.

Another factor that affects the pumping flow rate is the ID of the capillary tip. By controlling the parameters of the capillary puller, we fabricated pulled-tip capillaries with IDs of 30 μm, 50 μm, 60 μm, 70 μm, and 85 μm and measured their pumping flow rates under the same power input (4 V_pp_). As shown in Figure 3c, the flow rate increases as the tip ID increases. Capillaries with a large tip opening can generate more plumes per unit of time, which will translate to a high flow rate based on the mass conservation law. It should be noted that as the tip size increases, the minimum flow rate increases as well. Therefore, for applications requiring low flow rates, a smaller ID capillary is needed. In this work, the minimum flow rate achieved was ~3 μL/min with a 30 μm ID capillary. It is possible to achieve even lower flow rates with smaller ID capillaries. However, it is not recommended to use <30 μm ID capillaries for general applications due to the increased risk of tip clogging.

We also studied how the fluid viscosity affects the pumping flow rate. Similar to other fluid pumps, it is also difficult to pump high-viscosity fluids with the vibrating sharp-tip pump. As the fluid viscosity increases, it becomes difficult to atomize the fluid, leading to decreased pumping efficiency. We prepared liquid samples with different viscosities of 1.2 cP, 1.5 cP, 2.1, and 3.0 cP using glycerol. Figure 3d shows the relationship between the pumping flow rate and the fluid viscosities. As the fluid viscosity increased from 1.2 cP to 3.0 cP, the flow rate decreased from 54 μL/min to 23 μL/min.

We also examined the peak pressure that can be generated by the present pump. Since the 3D-printed device used in the above testing has a large cross-section, the actual pressure requirement is low. To test the capability of the present pumping method, we fabricated a PDMS microchannel with dimensions of 600 μm × 90 μm × 1 cm (width × height × length), as shown in Appendix A. By tracking the fluorescent particle velocity, the sharp-tip pump achieved flow rates ranging from 0.05 to 45 μL/min (Appendix A). Based on Poiseuille’s Law, the peak pressure generated by the sharp-tip pump is estimated to be ~0.2 kPa.

### 3.3. Multi-Pump Operations

The vibrating sharp-tip pump also allows the operation of multiple pumps simultaneously and independently. Here, we tested the operation of multiple vibrating sharp-tip capillary pumps with different reagents. For this study, we fabricated a microfluidic device with two inlets and two outlets, as shown in Figure 4a. Two vibrating sharp-tip capillary pumps with a tip ID of ~80 μm were used. The two pumps were connected to the two outlets of the microchannel. Both pumps were operated at their optimal frequencies of 94 and 95 kHz, respectively. To observe the fluid flow from the two pumps, the two inlets were connected to a Rhodamine B reservoir and a water reservoir, respectively. After turning on the two pumps simultaneously, we observed parallel streams of the fluorescent solution (Rhodamine B) and the water. When we increased the power input of the Rhodamine B pump, the flow rate ratio between the Rhodamine B and the water increased, indicating that the parallel streams were controlled by the two pumps. The total flow rate was controlled to be ~400 μL/min to show two streams clearly without extensive mixing. By carefully designing the fluidic path, it is possible to operate multiple pumps to deliver different reagents for chemical and biological analysis applications.

### 3.4. Performing Sequential Fluid Operations to Complete ELISA Protocol

Finally, we tested the vibrating sharp-tip capillary pump with complex fluid operations by performing a complete ELISA protocol. A complete ELISA protocol requires repeated cycles of reagent loading and washing. The vibrating sharp-tip capillary pump allows the convenient loading of different reagents by simply swapping the reagent reservoirs connected to the microchannel device. To perform the on-chip ELISA, we designed a 3D-printed microdevice with a center reaction chamber (Figure 5a). The center chamber holds the capture antibody-coated magnetic particles with a magnet. The suction pump was connected to the main outlet to provide fluid flow through the chamber. Two reagent inlets were employed for reagent loading and wash buffer loading, respectively. The reagent inlet was first connected to the sample solution and later switched to different reservoirs in the order of the detection antibody and substrate as the assay progressed, whereas the wash buffer inlet was connected to the wash buffer reservoir throughout the assay. Finally, to control the flow of the system, a plug valve was placed near the reagent inlet to ensure the chamber could be washed properly without cross-contamination. Here, we used a 1 ng/mL solution of Activin A standard as the target for the ELISA assay. The on-chip ELISA protocol is listed in Figure 5b. Once the capture antibody-coated beads were introduced into the well, the sample solution was pumped through the channel device. After an incubation period of 3 h, the washing buffer was pumped through the channel followed by the pumping of the detection antibody at a flow rate of μL/min. After 1 h of incubation, the beads were washed with ~1200 μL of ELISA washing buffer at a flow rate of 50 μL/min. Then, the QuantaRed HRP substrate was pumped into the reaction chamber. After 15 min of incubation, the fluorescence intensity was measured using an Olympus IX 73 epifluorescence microscope with proper filter cubes. When loading the reagents into the reaction chamber, the flow rate was set at 10 μL/min. During incubation, the flow was stopped. When the sample and detection antibody were washed out of the device, the flow was increased to 30 μL/min and 50 μL/min, respectively, to ensure the complete removal of the unbound reagents. As shown in Figure 5c, a clear difference in the fluorescence signals between the 1 ng/mL sample and the negative control sample was observed. This test showed the robustness of the present pumping method to perform complex fluid manipulations and operate over a long period of time. As the present pump is activated by RF signals, it will also be possible to achieve programmable fluid operations in the future.

## 4. Conclusions

We report a novel acoustic fluid pump based on the atomization phenomenon induced by a vibrating sharp-tip capillary. This pump is compatible with a wide range of microfluidic devices regardless of the channel material and fabrication methods. It is also simple and low-cost to make. No special fabrication is necessary, and each unit costs <$1. In addition, the low power consumption of the present method (~2–40 mW) makes it possible to be activated by simple signal generation and amplification circuits with battery power. Collectively, these features make this method a good candidate to perform complex and programmable fluid operations in point-of-care (POC) settings. To achieve the highest energy efficiency, we utilized the atomization phenomenon for fluid pumping. The potential issue with the atomization effect is the extra stress exerted on biological cells. Further studies are necessary to investigate whether the present method allows the recovery of cells with high viability and minimal stress. In future work, we will also explore the pumping phenomenon by inserting the vibrating sharp-tip capillary into a liquid to avoid the generation of aerosols and improve its biocompatibility.

## Figures and Tables

**Figure 1 micromachines-14-01212-f001:**
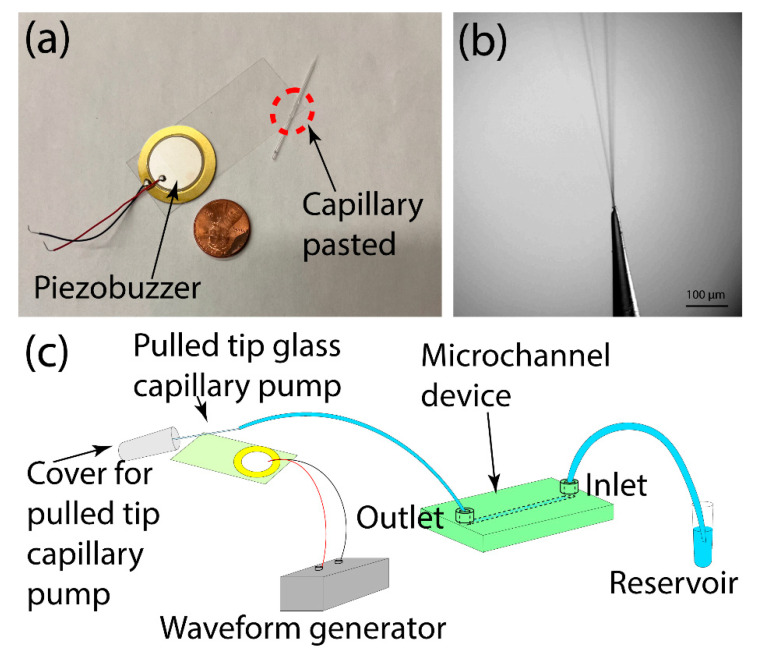
(**a**) A vibrating sharp-tip capillary pumping unit. (**b**) Ejection of fluid from the vibrating sharp-tip capillary. (**c**) Schematic of the vibrating sharp-tip capillary pump based on microfluidic system.

**Figure 2 micromachines-14-01212-f002:**
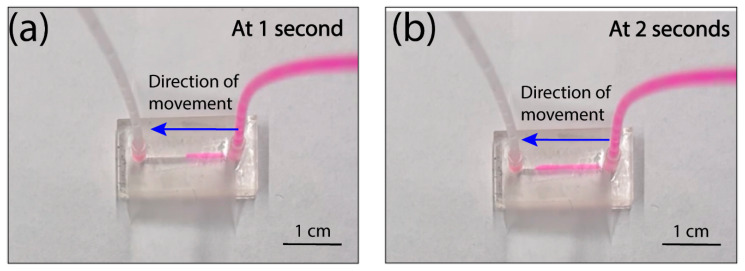
(**a**,**b**) Movement of the Rhodamine B solution inside the channel driven by the vibrating sharp-tip capillary pump.

**Figure 3 micromachines-14-01212-f003:**
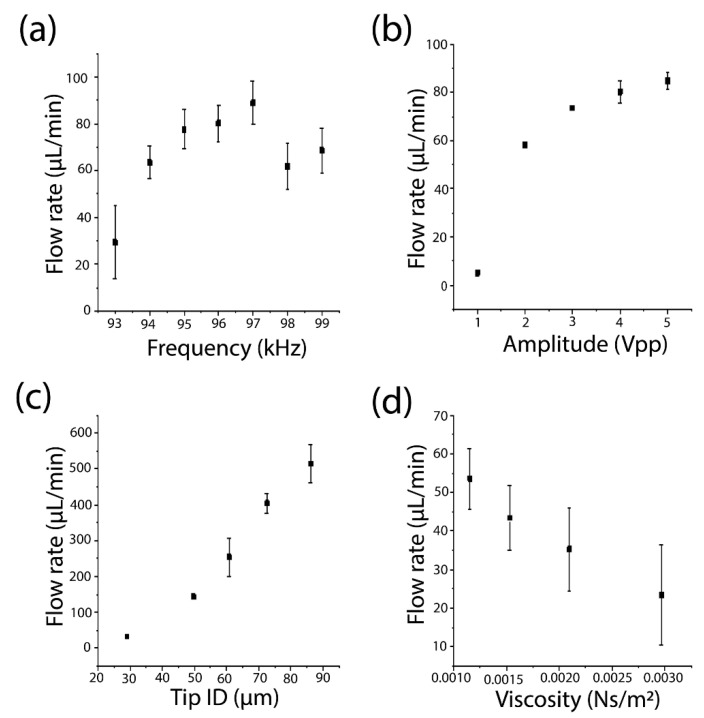
(**a**) Effect of input frequency in the range of 93 to 99 kHz on pumping flow rates; (**b**) Effect of input voltage on pumping flow rates; (**c**) Effect of the tip ID of the capillary on the flow rate; (**d**) Flow rate of solutions with different viscosities under the same input voltage and frequency. Error bars represent the standard deviation of three trials.

**Figure 4 micromachines-14-01212-f004:**
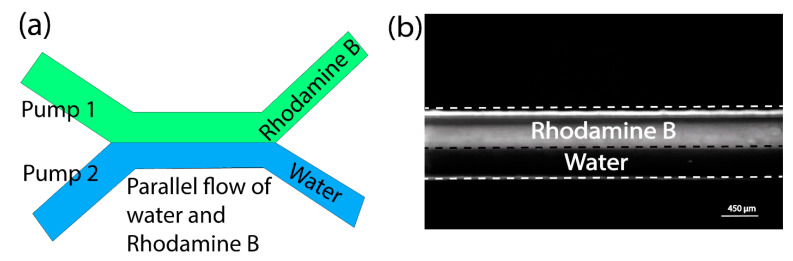
(**a**) Schematic of dual vibrating sharp-tip capillary pump set-up; (**b**) Microscopic view of parallel flow of Rhodamine B and water generated by the simultaneous operation of the two pumps.

**Figure 5 micromachines-14-01212-f005:**
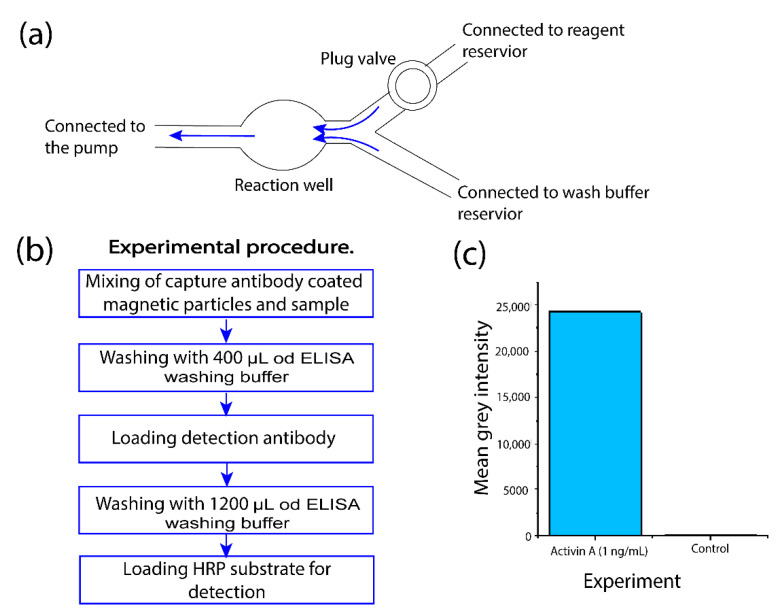
(**a**) Schematic of the experimental setup for the on-chip bead-based ELISA; (**b**) The workflow of the on-chip bead-based ELISA; (**c**) Fluorescent intensity of 1 ng/mL Activin A sample and a control sample.

## Data Availability

Data to support the conclusion of the manuscript are included in the manuscript. Additional data are available upon reasonable request to the corresponding author.

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
