# Peer review of "Acoustic Atomization-Induced Pumping Based on a Vibrating Sharp-Tip Capillary"

_micromachines, 2023, doi:10.3390/mi14061212_

Round 1

Reviewer 1 Report

The authors present a low cost, high flow rate controllable micropump system. The results here are very compelling, showing flow rates on the order of hundreds of µl/min with a very simple system with no moving parts. Further work is required before acceptance, however, where the pressure drop induced by this pump needs to be calculated, and the figures need to be improved (figure 5 is unreadable, for instance). This work should be considered for acceptance if the authors are able to address the following comments.

1.       What is the utility of a pump where the fluid needs to be atomized after it leaves a device? This would seem to preclude its use where the intention is to recover the sample. What fluids/systems would this approach not be appropriate for?

2.       In the process where the capillary is glued to the glass slide, is it important where the capillary is glued, what angle it’s glued and how far along its length the gluing occurs? Futher, figure 1 should be clearer in noting/labelling this gluing junction, as currently it looks as though it’s simply sitting on top of the slide.

3.       A critical characteristic of any pumping system is the pressure differential it is able to generate. I recommend that you calculate the pressure drop across the system (potentially as a second vertical axis of a different color for each of your plots in Figure 3), which can be done without any further experiments based on calculations considering your flow rate, the length of tubing and the channel dimensions. Your reference 22, for instance, examined the pressure drop generated by a SAW system, though achieved far low pressure differentials than what your system appears to be capable of. This is a far more relevant parameter than flow rate for your pump, as the flow rate will be determined by the configuration of the system just as much as it is the input parameters for your acoustic actuation. This is more similar, in this sense, to a pressure driven pump than a positive displacement pump (e.g. syringe pump).

4.       It would be worth comparing the present approach with other instances in which fluid was drawn through other materials and geometries based on acoustic actuation via atomization (e.g. doi.org/10.1103/PhysRevE.86.056312, doi.org/10.1016/0924-4247(96)80086-0, doi.org/10.1007/s10544-017-0152-9), and how the current setup using a capillary tube is advantageous for maximizing flow rate, ease of use, etc.

5.       The multi-pump flow description is very brief, where figure 4b presumable shows the flow when the pumps are operating at the same power. More data should be provided to support the statements given in lines 251-255. “it is possible operate multiple pumps to achieve complex fluid manipulations” – there are no ‘complex’ manipulations shown in this figure.

6.       Many figures are far too low quality for publication. Figure 1,2, 3 and 5 in particular need of improved resolution, where figure 5 is entirely unreadable.

7.       Figure 1b and Figure 2 could include a scale bar and labels. What does the atomized fluid look like after it’s ejected – how far does it travel and how wide is the spread? Another picture here could be useful.

Reviewer 2 Report

This manuscript introduces an intriguing method for pumping liquids into microchannels. It offers a novel approach, and the manuscript is generally well-constructed. However, one significant issue must be addressed.

The size of the device (microchannel) employed by the authors is extremely large, measuring 900 μm × 950 μm × 1500 μm (width × height × length). This far exceeds the dimensions that define a microchannel as shown in Table 1.1 of Teng's "Fluid Dynamics in Microchannels".

Let’s compare to other devices mentioned in the referenced literature. For example, the authors wrote “Tao et al. [15] reported a TSAW based pumping by applying RF signals to microfabricated interdigital transducers (IDTs). The device was modified with hydrophobic coating for improving the pumping performance. They were able to achieve pumping flow rate in the range of 0.1–0.2 μL/min with a power consumption of 2-7 W.” Although this flow rate appears much smaller than the 3–520 μL/min reported in this study, it's worth noting that the devices in reference 15 have dimensions of 250 μm × 70 μm and 500 μm × 70 μm (width × height).

Similarly, in another cited work - “Wu et al. [39] reported C shaped IDTs to generate localized acoustic streaming for fluid pumping. This method achieved flow rates in the range of 18.5 nL/min to 41.5 nL/min with a power consumption of 2-6 W.” The microchannel used in this reference measures 600 μm × 100 μm.

As one can imagine, the larger the microchannel, the easier it is to pump liquids. As a result, the significance and impact of the results obtained from such a large microchannel are questionable, and the comparisons made no sense.

If we apply the formulars (2-1) and (2-2) from “Fluid Dynamics in Microchannels”, we can infer that an increased device size results in a larger hydraulic diameter (Dh), which in turn leads to a significantly larger Reynolds number. This relationship is essential because the difficulty of pumping into microchannels is from low Reynolds numbers, where friction forces dominate and the liquid behaves as if it is more viscous, causing significantly large flow resistances. As a result, it's highly possible that the pumping results in the manuscript are primarily influenced by the oversized microchannel, and the atomization effect may not be as effective as readers anticipated.

Therefore, I recommend that the authors either fabricate a considerably smaller device (for example, 500 μm in width and 100 μm in height) and replicate the experiment, which would significantly enhance the paper's overall quality. Alternatively, the authors could revise the manuscript to highlight this size discrepancy when comparing their device to others and provide further discussions on the atomization effect.

There are some other questions that need to be addressed before it can be published.

1.       The resolution of all the figures is extremely low. In particular, the first few figures are difficult to read, while Figure 5 is completely unreadable.

2.       The authors should provide some explanation and discussion for the frequency dependency observed in Figure 3. There is no clear pattern for the distinct flow rates noted at varying frequencies, which is confusing.

3.       The authors' method of controlling the flow rate by adjusting the voltage amplitude applied to the piezoelectric device is great. However, it appears that what really matters is the vibration amplitude at the capillary tip. The process from voltage to tip’s vibration or atomization effect involves several steps - from converting the voltage to the vibration amplitude of the piezoelectric, to the subsequent vibration of the glass, and finally, the vibration of the tip. Could the authors provide an estimation of this conversion? The goal is to find a conversion between the applied voltage and the resulting atomization effect which would provide a more meaningful measure for comparing power consumption.

Round 2

Reviewer 1 Report

No further comments - this is innovative and interesting work.

Author Response

We thank the review's positive feedback. 

Reviewer 2 Report

Could the authors please provide additional information regarding the new experiment conducted on the newly fabricated device? For example, it would be helpful to have more details about the fabrication process of the PDMS microdevice, including optical images of the cross section. Additionally, could the authors demonstrate some details about the experiment that achieves a flow rate range of 0.5 to 40 μL/min?
